# A Conceptual Framework for Food Sharing as Collaborative Consumption

**DOI:** 10.3390/foods11101422

**Published:** 2022-05-13

**Authors:** Damayanti Octavia, Reza Ashari Nasution, Gatot Yudoko

**Affiliations:** 1School of Business Management ITB, Institut Teknologi Bandung, Bandung 40132, Indonesia; damayanti_octavia@sbm-itb.ac.id (D.O.); gatot@sbm-itb.ac.id (G.Y.); 2School of Economic and Business, Telkom University, Bandung 40257, Indonesia

**Keywords:** food waste, food sharing, collaborative consumption, sharing economy, social practice theory

## Abstract

Food waste has increased significantly and become a global issue amidst a growing concern regarding famine in several countries. Food sharing constitutes the solution to the problem provided an appropriate framework is developed that guides its application. The sharing economy was touted as the appropriate framework, yet it is excessively macroscopic to be able to capture the dynamics of food sharing activities. A microscopic framework is required to overcome this problem, the concept of collaborative consumption with its focus on activity level being one potential solution. However, an investigation into how food sharing activities can be viewed as collaborative consumption should be completed. This paper presents an analysis of the relationship between food sharing activities and collaborative consumption. The authors employed a systematic literature review conducted by meta-analysis and content analysis to identify the commonalities between the two and the theories underlying them. The result is a conceptual framework of food sharing activities as a collaborative consumption practice. The framework highlights eight propositions that can explain the intention, performance, and continuity of food sharing activities. At the end of the paper, the authors outline the theoretical and managerial contributions and recommend future research activities.

## 1. Introduction

Changes in food consumption patterns have increased the amount of food waste, which is the amount of edible food discarded for various reasons at successive stages of the food supply chain, such as excessive quantity and poor quality [1]. Food waste has risen to 32% of global food production, or 1.3 billion tons per year [2]. Worldwide, those primarily responsible for food waste are the retail sector, food service providers, and consumers (40%); households (25%); and agriculture (35%) [2,3].

Food waste negatively affects the environment, society, and the economy [4,5,6]. The impact of food waste on the environment lies in its contribution to greenhouse gas emissions (GHG) and global warming [7]. In terms of social consequences, food poverty is growing rapidly [8]. Finally, the economic impact is the loss of numerous resources such as water and energy [1,9].

Food sharing, an initiative to reduce food waste by redistributing surpluses [6,10], represents a response to United Nations (UN) Sustainable Development Goals (SDGs) 2 and 12 that relate to zero hunger and responsible consumption [11,12]. The practice of food sharing was initially restricted to household or kinship relationships such as family, neighbours, or friends. Technological developments have extended food sharing practices beyond sharing with relatives to include strangers. Moreover, such practices can accommodate large food industry or food services surpluses to be shared with the needy [3].

Previous food sharing studies have mainly employed the sharing economy as a fundamental sharing model [13,14,15]. However, a significant drawback of using the broad and macroscopic sharing economy concept is that it may fail to capture the dynamics of food sharing at the micro level [12,16,17,18,19], which is important because it encompasses the actors, their motivation, actions, and consequences of food sharing activities.

The concept of collaborative consumption is suitable for this purpose since it can specify the consumption pattern and context at a microscopic level in terms of both organizations and individuals [17,20,21,22,23]. The term “collaborative” refers to the cooperative interaction between the actors which builds a relationship of supportive trust and collective action [24,25]. Meanwhile, the term ‘consumption’ emphasizes sustainable utilization, which optimizes unused resources and reduces over-consumption by collecting and distributing food surpluses in line with food sharing goals [21,26,27,28].

Exploring food sharing from a collaborative consumption perspective is crucial. The micro perspective of collaborative consumption is appropriate to enhancing the effectiveness of food sharing as a means of reducing food waste because it can facilitate profound insights into food sharing activities, in particular the purpose, performance, and continuity of the activities [26,29]. From a scientific point of view, investigating food sharing activities from the viewpoint of collaborative consumption enriches the body of knowledge regarding such consumption, which has traditionally focused on the transportation, accommodation, and fashion sectors [10,30,31,32,33,34]. Of particular significance is the fact that food sharing activities involve non-monetary compensation. This is an important characteristic that differentiates food sharing from the other collaborative consumption activities. Hence, investigating food sharing activities by means of a collaborative consumption framework constitutes a substantial contribution to the literature.

Based on the elaboration in the previous paragraph, there are two objectives of this study. First, to recognize the characteristics of food sharing that are appropriate to the collaborative consumption context. Second, to develop a conceptual framework of food sharing from a collaborative consumption perspective. The structure of the study consists of four sections. The first introduces sharing as a new paradigm to reduce food waste, prompting the present research into food sharing within a collaborative context. The second addresses the research objectives by employing two methods: a systematic literature review and content analysis. The third section presents and discusses the results. The final section contains the conclusion and recommendations for further research.

## 2. Materials and Methods

Food sharing represents an embryonic issue and a minor topic that requires the developing of a solid understanding of previous studies through a literature review. In reducing food waste, many studies on food sharing are dominated by macro-level studies such as socio-ecological, geographic, economic, and cultural [35,36,37,38]. However, it is not optimal in reducing food waste because the main problem is at the micro-level, such as behavior at all levels in the food supply chain and improper food management [1,6,38,39]. Therefore, a critical approach is needed in food sharing research to deconstruct a new framework at the micro-level, which is reflected in the context of collaborative consumption. This study focuses on two things: the behavior of actors and the role of Information and Communications Technology (ICT) in food sharing activities.

This research is comprised of two stages in order to develop a conceptual framework of food sharing from the collaborative consumption perspective. Firstly, the study compares food sharing and collaborative consumption as a means of investigating the characteristics of and the relationship between both concepts using a meta-analysis. Secondly, the content analysis explores the first stage results as the fundamental inputs for conceptual framework development.

This study employs a systematic literature review in order to promote an understanding of food sharing within a collaborative consumption context. The review, conducted by both metaanalysis (using statistical analysis of R-studio Bibliometrix) and content analysis, develops a conceptual framework. A systematic literature review analyzes the transparency, methodology, and reproducibility of previous empirical research [39]. The transparency of the study methods employed is achieved through three stages: data collection, selection and extraction, and, finally, analysis. The proposed conceptual framework is reproducible through the application of both qualitative and quantitative methods.

Data collection involves the gathering of information from the database source and determining the keywords related to the study, namely, food sharing and collaborative consumption. The second stage consists of the selection and extraction of the relevant data. Finally, the R studio tool and content analysis address the research objectives. The successive stages of the research are described in Figure 1 below:
1Data CollectionThe study employed raw data from the Scopus database which, as the largest, most comprehensive, and robust database for citation, is the one preferred by the majority of scholars [40,41,42] The subsequent data collection stage identified the relevant keywords employed to select sample articles as the raw data [43], which was collected on 1 January 2022. Two keywords searched for separately in the titles and abstracts were “food sharing” and “collaborative consumption”. The study set a specific time frame of 2000 to 2022 when identifying the most recent food sharing and collaborative consumption studies. The initial search retrieved 296 articles related to food sharing and 113 on collaborative consumption.2Data SelectionThe study identified and removed irrelevant articles leading to the extracting of appropriate data from 41 articles on food sharing, and 13 covering collaborative consumption. The final data extraction consisted of 255 articles on food sharing and 100 articles concerning collaborative consumption for subsequent examination. The main “food sharing” and “collaborative consumption” data is shown in Table 1.The food sharing research period is more extended than collaborative consumption in the 2000s and the 2010s. The term “collaborative consumption” was first mentioned in a Felson and Spaeth article of 1978, but the term fell out of currency and was not employed again until the 2010s, when technology developed rapidly. Sharing activities using technology increased the volume of collaborative consumption research in the early 2010s. The 255 documents related to food sharing feature 692 authors, 742 author keywords, 11,335 references, and 183 sources. In comparison, 100 collaborative consumption documents contained 256 authors, 335 author keywords, 5583 references, and 65 sources.3Data AnalysisThe study has two research objectives. First, through meta-analysis, to recognize and map food sharing characteristics appropriate to a collaborative consumption context. A meta-analysis is a statistical method of integrating results from various studies as a means of identifying and comparing the patterns, disagreements, or relationships that appear in multiple studies on the same topic [39]. A meta-analysis employs Bibliometrix R-package tools which focus on keyword analysis, while Bibliometrix is an open-source software valuable for performing meta-analyses [44]. Secondly, the study’s objective is to develop a conceptual framework for food sharing from a collaborative consumption perspective through comprehensive content analysis and further research. The content analysis elaborates the thematic evolution of the discipline while drawing on the authors’ expertise [44].

## 3. Results and Discussion

### 3.1. Keywords Analysis

#### 3.1.1. Food Sharing

The keyword analysis describes the research trend, identifies the research gap and highlights future potential research [45]. Word Treemap contains the R-package Bibliometrix results that collate, extract and count the high-frequency keywords from the articles reviewed [46]. The Treemap reflects the characteristics of the field of studies and the production of concise and intuitive articles [47]. The study utilized keyword analysis in order to identify characteristics of food sharing and collaborative consumption studies. There are two types of keywords analysis. First, intensive keywords refer to keywords that are often used in the research topic. These keywords construct the core concept of the field. Secondly, episodic keywords mean that keywords are only mentioned one to three times in keyword analysis. The episodic keywords have a broad scope of context to describe a comprehensive trend of a research topic [48]. This study uses intensive and episodic keywords to convey a clear and comprehensive coverage. The first Treemap contains a keyword analysis of food sharing studies (see Figure 2 below).

There are fifty most prominent terms in food sharing studies. The terms are “food sharing” (19%), “food-sharing” (3%), “food waste” (7%), “sharing economy” (6%), and “sharing” (5%). Three terms have the same frequency (4%): “food”, “cooperation”, and “food security”. Food insecurity and sustainability (3%), “reciprocity”, “commensality”, and “exchange”, have the same percentage (2%). The last terms have a similar portion (1%): “reciprocal altruism”, “behavioural ecology”, “children”, “climate change”, “hunger-gatherers”, “social network analysis”, “social network”, “Canada”, “care”, “cities”, “consumption”, “Ethiopia”, “ethnography”, “evolution”, “food surplus”, “food system”, “health”, “hunger”, “indigenous”, “inequality”, “Inuit”, “motivation”, “resilience”, “360-degree video”, “activism”, “anthropology”, “Belgium, Berlin”, “coping strategies”, “costly signaling”, “COVID-19”, “cross-cultural comparison”, “cultural ecosystem services”, “culture”, “demography”, “dental caries”, and “diet”.

The fifty most frequently used keywords can be interpreted as follows:Of the fifty terms derived from the keyword analysis of food sharing studies, two duplicating terms: (1) “food sharing” and “food-sharing”; (2) “social networks” and “social network analysis” were identified.“Sharing Economy”, “COVID-19”, and “behavioural ecology” are recent issues in food sharing studies [9,12,49,50].“Food waste”, “food surplus”, “climate change”, “sustainability”, “consumption”, and “food system” imply behaviour to protect the environment [6,8,51,52].“Reciprocity”, “reciprocal altruism”, and “exchange” are the terms considered to explain food sharing motivation and are also similar to the basic principles of social exchange theory [29,49,50,53].“Motivation” is a keyword frequently investigated in food sharing research to understand why individuals participate in food sharing [51,52].“Sharing”, “food”, “cooperation”, and “care” reflect primary food sharing characteristics [54,55,56].The five terms refer to culture-related topics, namely “cross-cultural comparison“, “cultural ecosystem services“, “indigenous“, “Inuit“, and “culture“ itself [35,57,58,59].“Canada”, “Ethiopia”, “Belgium”, and “Berlin” refer to the country, region and city where food sharing studies are most widely conducted. The keyword “cities” indicates where most food sharing activities are undertaken in urban areas [60,61,62].“Children”, “hunger”, “inequality”, and “food insecurity” reflect the recipients in food sharing activities and indicate that food sharing studies emphasize creating food security and society’s well-being [36,63,64].Food sharing studies are concerned with “health” and “diet” [65,66].“Dental caries” have been excluded from the analysis because the keyword is not related to food sharing [67,68].“Social network analysis”, “ethnography”, and “360-degree video” are the tools and methods commonly used in food sharing research [63,64,69].“Hunter-gatherers” and “activism” are food sharing communities [37].“Demography” and “cost signaling” are other factors that influence food sharing research [70,71].

#### 3.1.2. Collaborative Consumption

Figure 3 contains the fifty terms most frequently used by the authors of collaborative consumption studies. “Collaborative consumption” itself is most frequently mentioned by the author (36%), followed by “sharing economy” (15%) and “sustainability” (4%). The other six terms: “trust”, “consumer behaviour”, “sharing”, “attitude”, “peer-to-peer”, and “sustainable consumption account” for 2%, while the rest each constitute 1%. There are 31 terms: “business model”, “constraints”, “materialism”, “theory of planned behaviour”, “access-based consumption”, “attitudes”, “business models”, “car sharing”, “carpooling”, “case study”, “collaborative consumption” (cc), “collaborative economy”, “community”, “consumer”, “customer-to-customer interaction”, “e-government”, “fashion sharing”, “Google Maps”, “intentions”, “mobile technology”, “motivation”, “motivations”, “netnography”, “online car-hailing”, “online collaborative consumption” (occ), “online food ordering”, “p2p”, “peer-to-peer”, “peer-to-peer sharing”, “perceived risk”, “perceived value”, “practice theory”, “product sharing”, “quality, service”, “social capital”, “social distance”, “social practice theory”, “traffic conditions”, “value”, and “values”.

However, of the fifty terms mentioned in the collaborative consumption analysis, the following keywords were identified:There are five sets of repeated terms, namely: (1) “collaborative consumption”, “collaborative consumption” (cc), and “online collaborative consumption”; (2) “peer-to-peer”, “p2p”, “peer-to-peer”, “peer-to-peer sharing”; (3) “motivation and motivations”; (4) “attitude and attitudes”; (5) “value and values” that are the most widely used in collaborative consumption research.Collaborative consumption research is dominated by two scopes: micro and meso. The microscope covers behavioural and interactional factors such as “trust”, “attitude”, “motivation”, “perceived risk”, “intention”, “perceived value”, “materialism”, and “social distance” [72,73,74,75,76,77,78,79,80]. Meanwhile, the meso-scope covers organizational factors such as “constraints”, “service, quality”, and “business models” [81,82,83,84].The most prominent characteristics of collaborative consumption are “peer-to-peer”, “sharing”, “customer-to-customer interaction”, and “mobile technology” [85,86,87,88,89,90,91].“Sustainability” and “sustainable consumption” are the two keywords indicating collaborative consumption studies of pro-environmental behaviour [92,93].In contrast to food sharing research, collaborative consumption research has been supported by several theoretical foundations, including the theory of planned behaviour, social practice theory, practice theory and social capital theory [18,94,95,96,97,98].Netnography and case study methods are commonly used in collaborative consumption research [27,99,100].The sharing economy, access-based consumption, and the collaborative economy intersect with collaborative consumption terminology [22,91,101,102,103,104,105,106].E-government, fashion sharing, online car-hailing, car sharing, carpooling, and online food ordering are the most frequently explored units of analysis in collaborative consumption [99,100,103,107,108,109,110]. Transportation has been widely discussed in collaborative consumption, while traffic conditions and Google Maps are considered in car-sharing [111].Customer and community are the actors of collaborative consumption activities [24,112,113,114].

#### 3.1.3. Food Sharing and Collaborative Consumption

This section divides the food sharing and collaborative consumption keyword analysis into six categories: characteristics, research objects, methods, theoretical background, unit analysis, and actors (Table 2).

CharacteristicsThere are similar characteristics between food sharing and collaborative consumption: sharing, sustainability, and the sharing economy; the latter of which is an umbrella concept for the sharing movement [16,114]. In most previous studies, the sharing economy and collaborative consumption are often defined. However, in others, collaborative consumption constitutes a subset of the sharing economy. It can, therefore, be interpreted that the sharing economy has a broader reach [12,16,115]. At the same time, the context of collaborative consumption is more specific. This finding supports our earlier statement on the scope of the sharing economy and collaborative consumption. Most food sharing research uses the sharing economy, rather than a collaborative consumption context, which means that food sharing studies fail to use this context in their microanalysis.Collaborative consumption is a sustainable business model which aims to create value from three factors: economic, social, and environmental [116], in line with food sharing activities that optimize food surplus to improve economic and social welfare and reduce food waste. Sharing is the primary activity of collaborative consumption and food sharing research. These similarities of food sharing and collaborative consumption characteristics indicate that future food sharing research can potentially be investigated within a collaborative consumption context. The differences between food sharing and collaborative consumption lie in the technology category and the application of the peer-to-peer system. Collaborative consumption is solidly in character with interactions between actors via the platform, while the prominent food sharing characters are those of “care” and “cooperation” (Figure 4).Research ObjectsCollaborative consumption involves a broader scope of sharing activity than in food sharing research, which focuses on food as the object. Transportation, fashion, e-government, and food are the objects of collaborative consumption research. Hospitality and accommodation also represent collaborative consumption research subjects not captured by the keyword analysis. Even though the word “food” appears in the collaborative consumption keyword analysis, the context is potluck, gathering, and online food delivery activities [92,109,111,117]. It means that the context of food sharing in optimizing food surplus remains limited [10,30,31,32,34]. The objective of collaborative consumption research is to investigate profit-oriented rather than non-profit models. Therefore, non-monetary or non-profit oriented collaborative consumption requires further investigation [114]. Food sharing forms part of the non-profit based sharing practice, together with commensality and social eating [118,119].MethodsQualitative methods dominate food sharing and collaborative consumption analysis. Collaborative consumption and food sharing analysis in the digital era is still at an early stage and, therefore, requires a more in-depth analysis. These two studies lack quantitative methods [120,121]. Quantitative methods support the validation of the qualitative results, primarily through netnography [122]. However, if explored more fully, quantitative methods are also widely employed. Both quantitative and qualitative methods have functions that mutually reinforce research results. The method employed depends on the research objectives to be achieved.Theoretical BackgroundReciprocal altruism and exchange are motivations highlighted in food sharing research. These two terms relate to Social Exchange Theory (SET) foundations. In SET, the interaction between two or more individuals constitutes a sustainable relationship if the benefits outweigh the costs that it entails. However, if the opposite is the case then the interaction will cease. Therefore, SET is more renowned as a cost and benefit analysis [123,124,125]. There are three categories of food sharing: for money, charity, and community [13]. Sharing for charity is a non-profit form of food sharing. Indeed, although it will not directly produce economic benefits, social and environmental ones will ensue [32].Three theories are captured in collaborative consumption research: the Theory of Planned Behaviour (TPB), Social Practice Theory (SPT), and Social Capital Theory (SCT). Several studies that use TPB as a theoretical basis are usually analysed using quantitative methods [77,110]. Most of the other articles combine TPB with other theories such as the Technological Acceptance Model, Self-Determination Theory, Norm Activation Theory, Value Belief Norm Theory, and Trait Theory [73,78,94,96,126]. All dimensions of TPB in combination represent theories that explore individual behaviour. SPT is the theoretical basis of qualitative analysis applied in several articles [127,128,129]. SPT has three dimensions: Meaning, Material, and Competence [127] which describe the practice of an organization. In contrast to TPB, which captures the individual aspect, SPT analyses the organizational role. As reflected in its multi-dimensionality, social capital theory also investigates structural, relational, and cognitive capital [98] from an organizational perspective.Unit of AnalysisFood sharing research is dominated by the macro context depicted by the sharing economy as the basis for more general and macro research, primarily into cultural and health issues that significantly influence the implementation of food sharing. Aspects of daily food consumption such as type of food, taste, and consumption patterns form part of the local culture. Food is also closely related to health since this is influenced by what is consumed [6,14,130]. Meanwhile, collaborative consumption research explores this issue more from the micro side. The behaviour of individuals or organizations is more widely discussed in the literature on collaborative consumption. Motivation, trust, attitude and intention represent the majority of research topics relating to collaborative consumption [18,73,78,94].ActorsBefore discussing food sharing actors, it is necessary to discuss food supply chain actors because there is a relationship between the two. Food supply chain actors consist of agriculture, manufacturing, wholesale, retail and the consumer [38]. Food waste can occur at all levels of the food supply chain. The factors causing food waste include: (1) the agriculture and manufacturing level: overproduction, aesthetic values, and storage [6,131]; (2) the wholesale and retail level: freshness, expiry date, aesthetics, poor management [1,38]; and (3) the consumer level: overconsumption, incorrect storage of food, meal planning, consumer behaviour [6,11,132]. The most preferred strategy for managing food waste is donating food to needy people [133]. The distribution of surplus food from the food supply chain is managed by food mediators such as food banks or food sharing communities that use digital applications. Therefore, it can be said that the food supply chain actor is part of the food sharing actor as a food contributor (Figure 4).Three actor roles are involved in collaborative consumption: the customer, the service provider, and the platform provider [85,134]. Food sharing also has three actors: donors, recipients, and mediators [36,93,135,136,137,138]. Food donors fulfil the role of providing services or products, which is the same as that of service providers. Meanwhile, food recipients are actors who receive products or services from service providers either directly or indirectly through food mediators. The latter act as intermediaries between food donors and food recipients whose role is almost the same as that of platform providers. The difference lies in using a technology platform and whether or not this food mediator can use it.

### 3.2. Problem-Solving Analysis

One of the units of analysis studied deals with the constraints faced in cases of collaborative consumption (Table 2). The research employs problem-solving analysis to address the second research objective of developing a food sharing conceptual framework within a collaborative consumption context. The problems encountered during the practice of food sharing are listed and described in Table 3. Explanation of these problems is essential to map and solve problems in the background environment. The conceptual framework in this study was developed against the background of the problems faced by the actors in food sharing (food donors, food sharing mediators, food recipients) in conformity with the research objectives of this paper.

Most of the obstacles that arise are due to a lack of experience with food sharing and limited familiarity with certain foods within a collaborative consumption context [10], especially the maturity of food sharing as a mediator. Therefore, the conceptual framework focuses on food sharing actors as mediators in this research. The theoretical basis of the food sharing conceptual framework employs social practice theory (SPT), which has three dimensions: competence, material, and meaning [146]. There are three reasons for adopting SPT as a theoretical framework in this research: Firstly, SPT can analyse alternatives when considering ‘problem’ behaviour which enables the accommodating of problems that arise in food sharing practice [147]. Secondly, most studies that use SPT as a theoretical framework analyse sustainable consumption in line with the goal of food sharing within a collaborative consumption context [147,148,149]. Thirdly, practice is the smallest unit of analysis that can form a pattern of social practice on the basis of individual behaviour. The use of SPT in developing a food sharing conceptual framework can investigate in microanalysis, which is still rarely studied in food sharing research [147].

The efforts to reduce the obstacles encountered in food sharing activities are contained in Table 4. The problem-solving indicators in the table can be used as dependent ones and the impact of problem-solving in reducing the constraints can be used as an independent indicator. The problem-solving analysis meets the criteria of rigor and exactness, parsimony, and logical consistency in creating the quality of indicators [150].

The contents of Table 5 form the basis of reference and identify indicators for designing a food sharing research framework from a social practice theory perspective. The findings of the problem-solving analysis are explained in the following sub-chapter in the form of a comprehensive food sharing framework.

### 3.3. Conceptual Framework

This conceptual framework is structured on the basis of the four stages of Dubin’s theory-building method [154]: (1) Unit: identifies the basic theoretical concepts; (2) Law of interaction: explains the interaction and interrelationships between theoretical units; (3) Boundaries: defines the boundaries of the conceptual framework; (4) System states: the condition of the system in which the theoretical units interact and operate differently (Figure 5).

1Units of AnalysisThe unit analysis of this study employs the three dimensions of SPT proposed by Shove (2012) and the extended attitude dimension from the Theory of Reasoned Action (TRA). The attitude as an extended dimension assists in clarifying the impacts of the three dimensions of social practice. Attitudes refer to an individual perceiving specific behaviour as favourable or unfavourable. Intentions, in turn, are assumed to capture the motivational factors that influence such behaviour [74]. Practice theory covers the dynamics of daily life and what people actually do [155]. The three elements of social practice theory are meaning, material, and competence [146]. Meaning refers to the interpretation, perspective, and characteristics of the practices. Material constitutes all the tangible elements that support practices, including technologies, tools, and hardware. The material can also be divided into three topics: infrastructure, device, and resources. Competence is defined as the expertise, understanding, skill, and knowledge necessary to implement the practice [155,156,157].2Law of InteractionThis law covers the combination of the three components of social practice theory (meaning, competence and material) in a unified whole, interrelated and supporting one another to implement a social practice [146].3BoundariesThere are three models of food sharing: for profit, for the community, and for charity [3]. This proposed framework is limited to non-monetary collaborative consumption that is specifically related to food sharing for charity. The previous research into collaborative consumption generally discussed fashion, accommodation, and transportation as producing monetary compensation, while devoting less attention to food sharing as non-monetary collaborative consumption [10,30,31,32,33,34,109]. The conceptual framework focused on the food sharing mediator that affects the mediator and other actors, such as food donors and food recipients.4System StatesThe system states of this proposed framework include food sharing in the urban area. Food sharing communities are generally established in conurbations for three reasons. First, inhabitants of rural areas tend to be more satisfied with simple plant-based dishes, while city dwellers enjoy delicious food [135]. This fact, allied with the behaviour of urban communities and various other reasons [143], causes more food waste in urban areas. Second, the major characteristic of urban communities is individualistic, which results in social isolation. Consequently, food sharing is necessary in urban areas to promote social inclusion [135,151]. Third, the high incidence of poverty and hunger in urban areas necessitates food sharing in order to reach those individuals requiring food [158]. Besides urban areas, food sharing constitutes an increasing phenomenon in the global socio-economic crisis [159].

The analysis of problem-solving, the meaning of food sharing, competencies required by food sharing mediators, as well as the tools and materials that can support all activities carried out by food sharing mediators can be generalized and described as follows (Figure 6):

The three dimensions of social practice theory are described through the indicators resulting from the content analysis. First, the meaning dimension: food sharing mediators need to re-position their values through three sustainability values, namely; economic, social, and environmental in order reduce the stigma of shame on the part of food recipients who receive assistance. Second, the competence dimension: skill and capabilities in the areas of technology and food management increase the trust of food recipients in the quality of food distribution and optimize the role of ICT in enhancing the performance of food sharing mediators. Knowledge of marketing communications for non-profit organizations provides the information required by food donors to increase food sharing participation and to educate the public to improve sustainable consumption behaviour. Third, the material dimension tangibly supports food sharing practice. The prerequisite materials of food sharing practice comprise the ICT infrastructure to increase food sharing performance; storage infrastructure to maintain the quality of perishable food; appropriate transportation to cover long distances and accelerated food donation distribution to prevent their accumulation and from becoming food waste in storage areas; modifying the distribution channel through cooperating with local communities to distribute food donations that are difficult to access by food sharing mediators; and a legal tool to increase participants’ trust in food sharing mediators.

Figure 6 shows that ICT is an essential factor in food sharing and handling food waste. There are three crucial roles of ICT in reducing food waste [38], First, digital platform technologies serve as a liaison between food donors and recipients and can expand the food stakeholder network. The digital platform is also an interactive and responsive communication tool, making it easier for all food-sharing actors to interact [160]. Second, RFID (Radio Frequency Identification) is beneficial for more controlled inventory management, more structured food category management and better food layout management. Third, monitoring food quality, humidity, and temperature according to the type of food and managing expiration dates can use the IoT (Internet of Things).

From the content of Figure 5, it can be concluded that social practice dimensions affect the level of trust of food recipients and donors, sustainable consumption behaviour, the performance of food sharing mediators, and the intention to participate in food sharing activities, as depicted in Figure 7. Based on the framework it contains, there are eight propositions: Proposition 1: Meaning, competencies, and materials of food sharing mediators engender the trust of food donors and food recipients [76,126,161].Proposition 2: Meaning, competencies, and materials influence sustainable consumption behaviour [162].Proposition 3: Meaning, competencies, and materials improve the performance of food sharing mediators [81].Proposition 4: Trust influences food donors’ intentions and food recipients’ participation [82].Proposition 5: Sustainable consumption behaviour intensifies commitment to food sharing participation [25,97].Proposition 6: Food sharing mediators’ performance affects the intention of food donors and food recipients to participate [92].Proposition 7: Food sharing mediators’ performance increases the trust of food donors and food recipients [92].Proposition 8: Meaning, competencies, and materials of food sharing mediators influences the intention to participate [28].

These eight propositions are theoretical contributions for further research. The framework in this research can be analyzed using qualitative and quantitative methods to obtain empirical results that can improve research findings.

## 4. Conclusions

This research has two objectives. The first is to recognize the characteristics of food sharing that are appropriate to a collaborative consumption context. The second is to develop a conceptual framework for food sharing within a collaborative consumption context. The first result is that the characteristics of food sharing emphasize the social aspects of care and cooperation, while collaborative consumption prioritizes the use of technology platforms and peer-to-peer interactions. The similarities between the characteristics of food sharing and collaborative consumption is that they have the goal of sustainability, sharing activities, and the sharing economy as their umbrella concept. This similarity of characteristics is used to develop a food sharing conceptual framework from a micro-analysis perspective within the context of collaborative consumption. Previous research has discussed many macro-levels such as socio-ecological, economic and cultural. However, food waste is still high because the main problems are at the micro-level such as consumer behaviour and food handling management. Therefore, this study focuses on micro-units of analysis to optimize strategies for reducing food waste through food sharing. Secondly, indicators for each dimension of social practice theory in food sharing practice and eight propositions from the conceptual framework developed in the theory of social practice perspective were identified.

This study has several limitations, the first being that the research framework constructed in this study only covers food-sharing mediators. At the same time, there are two other actors in food sharing activities: food donors and food recipients. Therefore, further research is encouraged to develop a social practice conceptual framework for food donors and food recipients because the meaning, competence, and materials will be different for those actors [163]. Secondly, the research constitutes a literature review analysis. Therefore, an empirical analysis involving qualitative or quantitative methods, which are more relevant, is required to clarify the conceptual framework. Thirdly, research on food sharing is predominantly carried out in developed countries [61,62]. Therefore, further research can be carried out in developing countries to compare and improve social practice within food-sharing contexts [156]. Fourthly, there are three models of food sharing: food sharing for profit, for charity, and for community [3]. This study develops a conceptual framework of food sharing for charity. Future research into the social practice framework relating to food sharing for profit and for community can be conducted.

## Figures and Tables

**Figure 1 foods-11-01422-f001:**
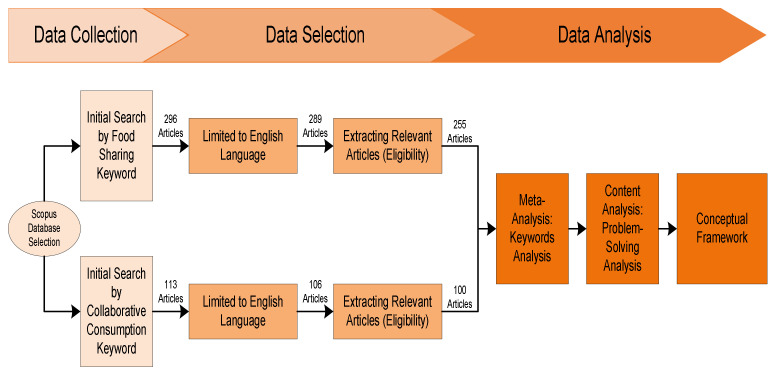
Research method.

**Figure 2 foods-11-01422-f002:**
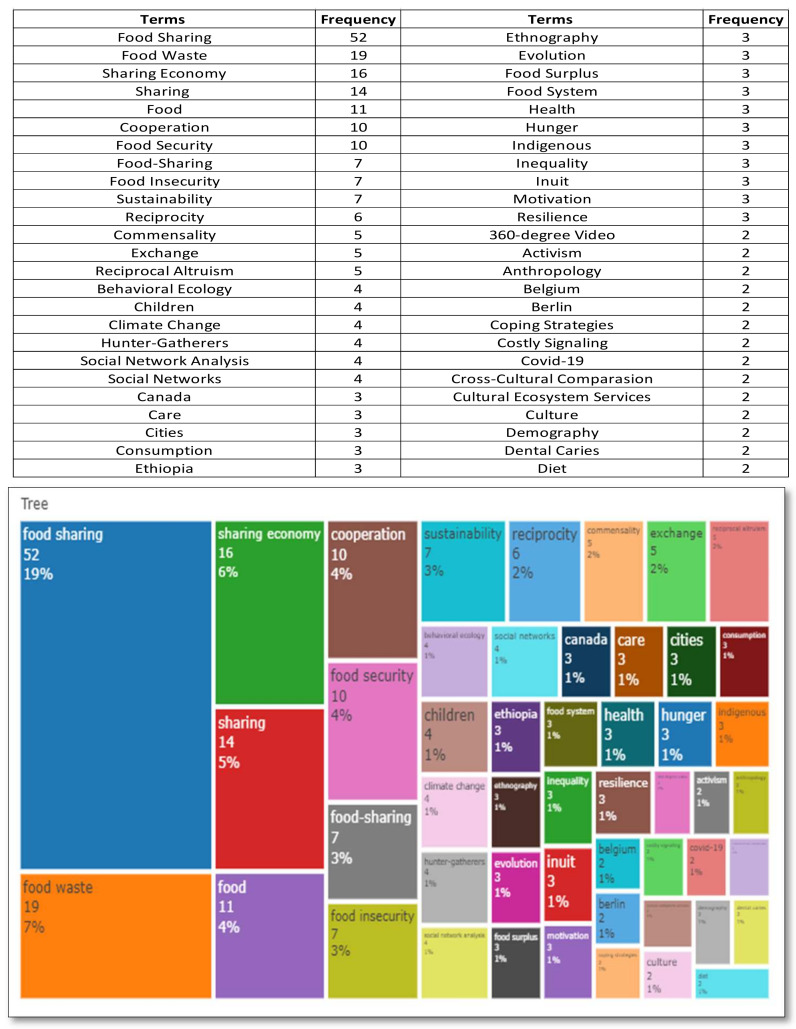
The food sharing treemap.

**Figure 3 foods-11-01422-f003:**
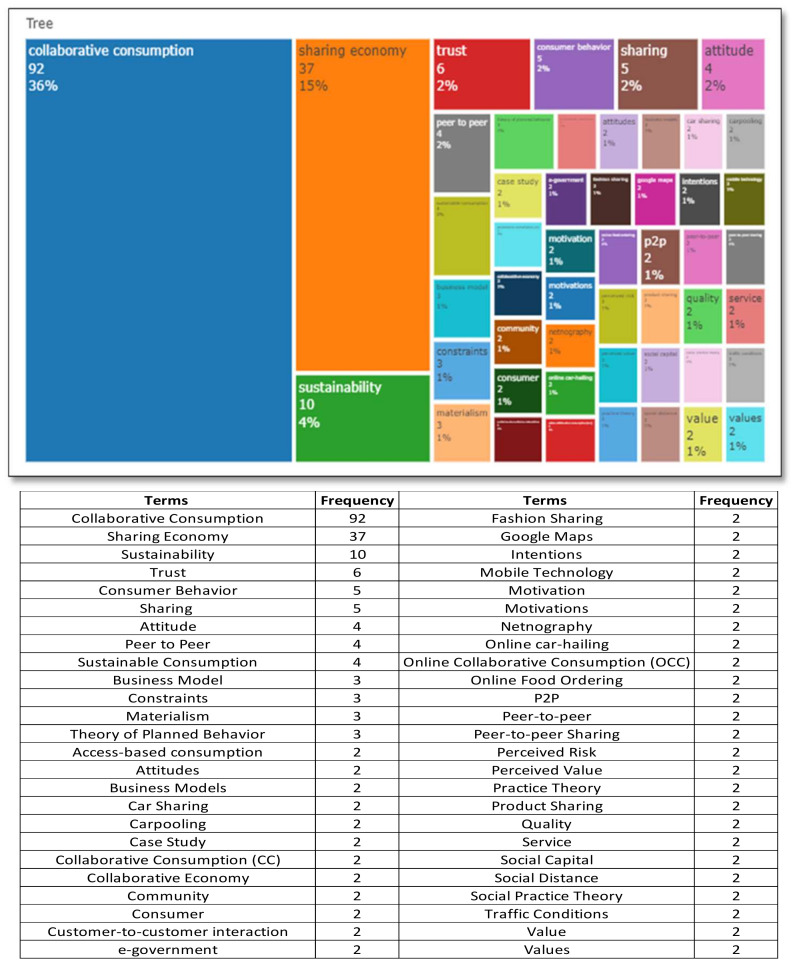
The collaborative consumption treemap. Source: R-Studio for Bibliometrix Tools.

**Figure 4 foods-11-01422-f004:**
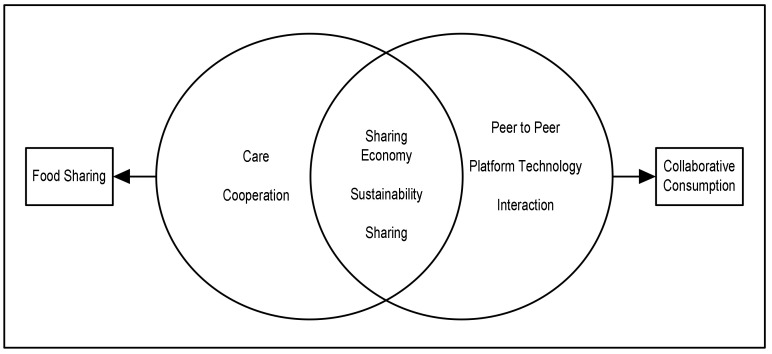
The Characteristics of food sharing and collaborative consumption.

**Figure 5 foods-11-01422-f005:**
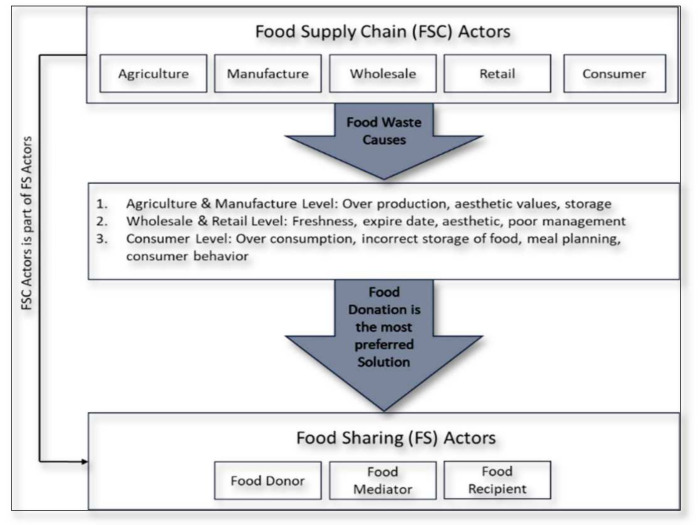
The relationship between FSC actors and FS actors.

**Figure 6 foods-11-01422-f006:**
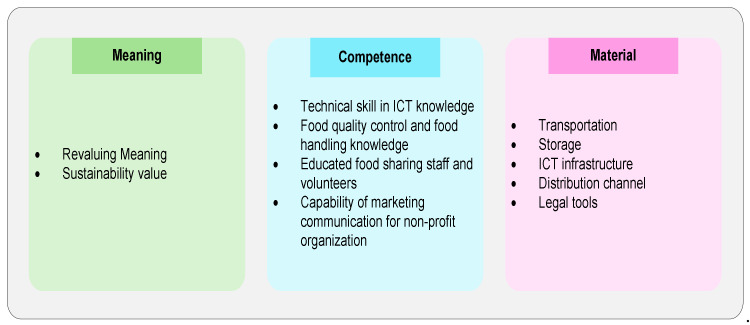
Food Sharing from the perspective of Social Practice Theory.

**Figure 7 foods-11-01422-f007:**
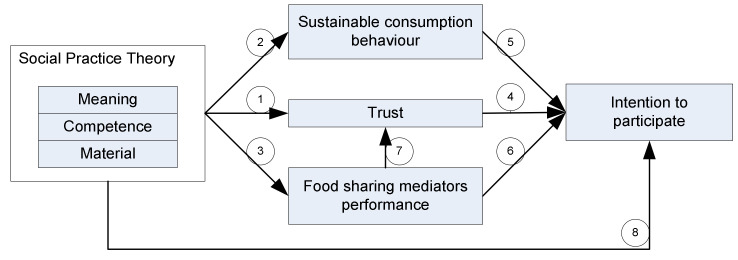
A conceptual framework for food sharing as collaborative consumption.

**Table 1 foods-11-01422-t001:** Main data.

Descriptions	Food Sharing	Collaborative Consumption
Timespan	2000:2022	2013:2021
Sources	183	65
Documents	255	100
References	11,335	5583
Authors	692	256
Author’s Keywords	742	335

Source: R-studio Tool.

**Table 2 foods-11-01422-t002:** The description of food sharing and collaborative consumption research.

Description	Food Sharing	Collaborative Consumption
characteristics	Sharing Sustainability Sharing Economy Cooperation Care	Peer-to-peer Platform Technology Sharing Sustainability Sharing Economy Interaction
research objects	Food	Transportation Fashion E-Government Food
methods	Ethnography Social Networks Analysis	Netnography Case Study
theoretical background	Social Exchange Theory	Theory of Planned Behaviour Social Practice Theory Social Capital Theory
unit of analysis	Micro: Motivation Macro: Demography, Culture, Region, Health	Micro: trust, attitude, motivation, perceived risk, intention, perceived value, materialism, and social distance Meso: business model, service, quality, and constraints
actors	Food recipients Community: food donors and food mediators	Customer Service provider Platform provider

**Table 3 foods-11-01422-t003:** Problems Analysis.

Actors	Problems
Food Donors (FD)	Lack of information about and awareness of available food-sharing alternatives [10]—FD1; Insufficient amounts of food available to donate through food sharing [139,140]—FD2; Food sharing locations that are difficult to access [140]—FD3; Limited capacity to recycle food waste [30]—FD4; Insufficient time to collect excess food [140]—FD5.
Food Sharing Mediators (FM)	Food distribution is subject to a time limit due to perishable food being close to its expiry date and cooked food having to be used before it is no longer edible [141]—FM1; Limited distance range of food distribution networks [6]—FM2; Low interest in food sharing platforms whose content would improve the behavioural response of food sharing service users [15]—FM3; The use of digital platforms promoting food sharing activities is still not optimal [142]—FM4; Low skill levels and limited training provision [116]—FM5; Limited resources: ICT infrastructure [99]—FM6; Proper transportation and storage infrastructure require improvement to reduce the amount of food waste [6]—FM7.
Food Recipients (FR)	The dependence of recipients on the food sharing community [9]—FR1; The stigma that food recipients experience when receiving food assistance [10,143,144]—FR2; Food recipients’ lack of trust in the quality and hygiene of donated food provided by food mediators [10,82,145]—FR3.

**Table 4 foods-11-01422-t004:** Problems-solving analysis from a Social Practice Theory perspective.

	Problem-Solving Indicators	Code of the Problem Solved
Meaning	Transform and re-position value from commodity to gift, from poverty to prosperity, from waste to conservation [8,144,151]; Social value: food sharing creates a social space in which to meet new people and demonstrate caring behaviour [3,56]; Environmental value: food sharing reduces food waste [8]; Economic value: improving CSR value and business sustainability for food donors [151], reducing spending on food [152].	FR2
Competence	Technical skill to optimize ICT function [6]; Ability to assess the quality of food donations [141]; Education and training [121]; Communication capability through creative campaigns [141], WOM and publicity [14], events [8], social media marketing [153] to provide food sharing information and education.	FD1–4 FM3–5, FR1–3
Material	Appropriate transportation [6]; Storage infrastructure [6]; ICT infrastructure [99]; Legal tools [62]; Distribution channel modifications including suitable options and allocation means that the utilization of surplus food can be drastically improved [6]; Providing mobile apps to increase familiarity [15].	FD3 & FD5 FM1–2 FM6–7 FR1

CSR: Corporate Social Responsibility; ICT: Information and Communications Technology; WOM: Word of Mouth.

**Table 5 foods-11-01422-t005:** Conceptual Development Process.

No	Process	Description
1	Units	Meaning, Competence, Material, Attitude
2	Law of interaction	Interrelated units
3	Boundaries	Food sharing for charity
4	System states	Urban area

## Data Availability

The study did not report any data.

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
