# Peer review of "A Conceptual Framework for Food Sharing as Collaborative Consumption"

_foods, 2022, doi:10.3390/foods11101422_

Round 1

Reviewer 1 Report

The paper “A Conceptual Framework for Food Sharing as Collaborative Consumption” deals with very interesting and important topic. The article has a theoretical character. The issues related to the food waste are still important and up-to-date. And they are especially important when they are considered from the perspective of the food consumer, i.e. each of us.

I have a few minor comments to correct in the text of the manuscript:

In the text, reference numbers should be placed in square brackets [1–3] or [1,3]. Please correct citation - line 31.

Figures 2 and 3 are unreadable, please correct them.

I suggest mentioning the sources of the presented data under the tables and figures.

Overall, it is a well-written manuscript and it has been a pleasure reading it.

Author Response

Dear Reviewer 1,

We greatly appreciate your time and effort in reviewing this manuscript. We found your comments helpful in refining the paper. We have prepared our responses to your valuable comments as follows:

Comment 1

In the text, reference numbers should be placed in square brackets [1–3] or [1,3]. Please correct citation - line 31.

Response 1

We have corrected the reference to square brackets in the following sentence on line 31 of the paper:

Food waste has risen to 32% of global food production, or 1.3 billion tons per year [2].

Comment 2

Figures 2 and 3 are unreadable, please correct them.

Response 2

The revision is adding the tables below figures 2 and 3 to make them readable. I hope it is sufficient to address this comment as well.

Comment 3

I suggest mentioning the sources of the presented data under the tables and figures.

Response 3

Sources have been added for tables 1, figures 2 and 3. The other figures and tables are the author's own analysis.

Reviewer 2 Report

I thank the editor for giving me the opportunity to review this interesting paper dealing with food waste and sharing practices to reduce it. The paper is of interest to Foods’ readers, but I have some concerns:

1) Sharing activities to reduce food waste should be discussed analyzing the different food supply chain actors: input supplier, farmers, producers/processors, retailers and consumers.

2) Authors may want to deeper the discussion on the role of Information  Communication Technologies introduced in the text and explain in detail the mechanism ICT may contribute to lower food waste within each actors of supply chain as well as along the entire food supply chain.

3) Food supply chain actors may implement food sharing according to their geographical location due to cultural variation across countries. Authors may want to discuss this point.

Concluding, the authors introduce the concept of food sharing as a collaborative practice to lower food waste, but they did not thoroughly discuss this topic in the text, as well as did not use a critical approach.

Author Response

Dear Reviewer 2,

We greatly appreciate your time and effort in reviewing this manuscript. We found your comments helpful in refining the paper. We have prepared some responses to your constructive comments as follows:

Comment 1

Sharing activities to reduce food waste should be discussed analyzing the different food supply chain actors: input suppliers, farmers, producers/processors, retailers and consumers.

Response 1

We have added picture 4 and an explanation on lines 370-382 regarding the relationship between food supply chain actors and actors in food sharing in the picture.

Comment 2

Authors may want to deeper the discussion on the role of Information Communication Technologies introduced in the text and explain in detail the mechanism ICT may contribute to lower food waste within each actors of supply chain as well as along the entire food supply chain.

Response 2

We have added the role of ICT in reducing food waste on lines 512-520.

Comment 3

Food supply chain actors may implement food sharing according to their geographical location due to cultural variation across countries. Authors may want to discuss this point.

Response 3

Thank you for your suggestion. Cultural and geographical studies are widely used in food sharing, but the number of food sharing is still high because the main problem lies at the micro-level such as the behaviour of all actors in the food supply chain and food handling management, so this research focus on micro-level analysis. We will provide an explanation of this on lines 559-563.

Comment 4

the authors introduce the concept of food sharing as a collaborative practice to lower food waste, but they did not thoroughly discuss this topic in the text, as well as did not use a critical approach.

Response 4

Thank you for your suggestion. We have added the relevant discussion on lines 81-89.

Round 2

Reviewer 2 Report

I am currently fine with the current version of the manuscript.

Author Response

Thank you for your appreciation.